# Spatial Effects of Air Pollution on the Siting of Enterprises: Evidence from China

**DOI:** 10.3390/ijerph192114484

**Published:** 2022-11-04

**Authors:** Xuna Zhang, Shijing Nan, Shanbing Lu, Minna Wang

**Affiliations:** The School of Economics and Management, Northwest University, Xi’an 710127, China

**Keywords:** air pollution, the siting of enterprises, spatial spillover effects, mechanism analysis

## Abstract

The siting of enterprises is important for enterprises to formulate business objectives and business strategies, both of which are crucial to the development of enterprises in the future. Although there exists an irrefutable fact that the increasingly serious environmental problems are affecting the behaviors of enterprises, how air pollution affects the siting of enterprises has received little academic attention. Therefore, using the dataset of Chinese prefecture-level cities from 2014 to 2020, this paper employs the Spatial Durbin Model to investigate the direct and spatial spillover effects of air pollution on the site selection of enterprises. In addition, this paper also establishes a mediation effect model to explore the impact mechanism of air pollution on the site selection of enterprises. The empirical results show that air pollution exerts a negative impact on both the local and spatially related regions’ enterprises’ site selection, and the above conclusion is reinforced through a series of robustness checks. The heterogeneity analysis demonstrates that air pollution has a greater inhibitory effect on the siting of low-cleaning enterprises and small-scale enterprises for the local and adjacent regions. The mechanism analysis results indicate that air pollution inhibits the siting of enterprises by reducing the local labor endowment and market scale. Our study enriches the relevant theory of air pollution and enterprises’ location nexus, and it also provides an empirical basis for the Chinese government to formulate policies related to air governance and the siting of enterprises.

## 1. Introduction

In recent years, the problem of air pollution has attracted worldwide attention. Serious air pollution has brought great pressure on people’s lives and health. As the World Health Organization stated, at least one million people around the world died prematurely due to air pollution, the bad situation would continue for a long time and was likely to get worse [1]. The site selection of enterprises, regarded as the main body of the market economy, directly affects the level of economic development and environmental quality in each city. However, how air pollution affects the site selection of enterprises has received little attention.

Air pollution has been demonstrated to have causal effects on a range of health and economic aspects: infant mortality, cognitive, lung function, mental health, income inequality, labor productivity, food production, labor market decisions, and health expenditures [2,3,4,5,6,7,8]. These results suggest that air pollution worsens the living conditions of enterprises. Firstly, air pollution damages human health directly, cuts down people’s working hours and decreases their work efficiency, and, to reduce the damage of air pollution, employees even prefer to migrate to places with suitable air quality [9], all of which can reduce enterprises’ productivity [10]. Secondly, air pollution increases residents’ medical expenditure, worsens residents’ expectations for the future, and reduces their propensity to consume [11]. As a result, a city with severe air pollution will reduce its labor endowment and market scale. The shortage of labor and the reduction of sales volume further undermines enterprises’ productivity and competitiveness [12,13], which may force enterprises to rethink their site selection. Some scholars introduced the environmental factor into the new economic geography model and found that environmental pollution inhibited the site selection of enterprises and became the dispersion force to reduce economic agglomeration [14].

A bulk of the literature has focused on the negative effects of air pollution; however, to the best of the authors’ knowledge, none of them have clearly answered the question about whether and how enterprises change their location in response to air pollution. The only existing relevant literature has not clarified the impact mechanism of air pollution on the siting of enterprises. It is apparent that enterprises probably relocate to pleasant air-quality regions seeking higher productivity, which is analogous to labor migration. Optimizing the layout of enterprises is the key to improving the core competitiveness of the region, and also an important thrust to achieve high-quality economic development. Therefore, it is of great practical significance to figure out correctly the influence of air pollution on enterprises’ site selection. Another shortfall in the existing research is that it has not taken into account the spatial correlation when investigating the effects of air pollution on enterprises’ site selection. Spatial correlation means the potential interdependence between a phenomenon and other phenomena around it. The traditional non-spatial panel data models result in biased estimations when the spatial correlations exist [15,16].

The siting of enterprises is closely related to the development of a region. Investigating the relationship between air pollution and enterprise site selection will help people to understand correctly the damage caused by air pollution to the regional economy. Therefore, this paper takes Chinese prefecture-level cities as the research samples and adopts spatial econometric techniques to explore how air pollution affect enterprises’ site selection and the impact mechanism. Our study contributes to the existing literature from two perspectives. Firstly, from the perspective of environment, this research investigates the impact mechanism of air pollution on enterprises’ site selection and proposes two influence channels of labor endowment and market scale, breaking through the previous research framework of enterprises’ site selection and enriching the enterprises’ location theory. Secondly, based on the spatial panel econometric model, the influences of air pollution on enterprises’ site selection are researched all around, and the spatial spillover effects of air pollution on enterprises’ siting are identified. There are a few studies investigating the effects of air pollution on enterprises’ site selection; however, none of them employ the spatial panel data model as far as we know. Using this model offers certain benefits by incorporating the spatial interaction effects. Omitting the spatial effects may cause distortions in the parameter estimates.

The remainder of this paper is arranged as follows. Section 2 summarizes relate literature. Section 3 introduces the data and methods. Section 4 presents and discusses the empirical results. Section 5 concludes the main findings and puts forward several policy proposals.

## 2. Literature Review and Research Hypothesis

### 2.1. Impact Mechanism of Air Pollution on Enterprise Site Selection

The site selection of enterprises refers to how to use scientific methods to determine the geographical location of enterprises, so that it can be organically integrated with the overall operation system of the enterprises, and thus allowing the business purpose of the enterprises to be achieved effectively and economically. Enterprises often consider the constraints of target location factors and their own characteristics, and try to find a spatial suitable location that can make it easier to achieve their final goals. Enterprises’ site selection includes location selection of new enterprises, reconstruction, expansion or relocation of existing enterprises. A large number of studies have investigated the determinant factors for enterprises’ site selection [14,17,18,19], which provides a valuable reference for our study. Early studies consider that regional economic conditions such as labor endowment, market scale, tax policy, labor wage and infrastructure are the main impact factors for enterprises’ site selection [17,18,19]. Some scholars also believe that regional culture and environmental regulations play an important role in enterprises’ site selection decisions [20]. The latest research mainly analyzes the impacts of enterprises’ characteristics such as enterprises ownership, scale and industry classification on enterprises’ location decisions [14].

Although much of the literature has studied the influencing factors of enterprises’ location, little of the literature has explored the impact of air pollution on enterprises’ site selection. Through systematic review of relevant literature, it can be concluded that air pollution mainly affects the site selection of enterprises by reducing labor endowment and the shrinking market scale.

Some researchers point out that long-term exposure to air pollution significantly increases the risk of cardiovascular diseases, such as lung disease, heart disease and hypertension [21]. Meanwhile, air pollution can cause respiratory diseases and increase stroke mortality [12,22]. Because the elderly and infants are more vulnerable to the effects of air pollution, the labor force may need to spend more energy and time on taking care of their families, which will result in shorter labor supply time. In addition to the physical impact on physical health, air pollution has also been found to have significant negative effects on human mental health [23,24]. For example, anxiety symptoms are severe and increase the chance of depressive symptoms. Such mental health problems may reduce an individual’s life satisfaction and shorten an individual’s life expectancy. The theory of population migration points out that a bad living environment forces people to leave their hometown, while good expectations bring people to a new place [25,26]. Thus, air pollution decreases the labor endowment of a region. According to the industrial location theory, enterprises’ site selection depends on the stock of production factors such as labor force, capital, and technology to a large extent. Labor and capital constitute the main body of productive input of enterprises and play a decisive role in the productive output of enterprises [27,28]. It is well known that labor force is an important factor in the site selection of enterprises. Many enterprises have been forced to shut down due to labor shortages. The lack of a labor force reduces the matching between employees and enterprises. Enterprises will transfer with the reduction of local labor endowment. Thus we can conclude that air pollution inhibits the site selection of enterprises by reducing labor endowment.

Market scale is often related to purchasing power. Air pollution reduces purchasing power in two ways. On the one hand, air pollution increases income inequality to reduce the purchasing power of the market. As mentioned above, air pollution increases the health burden of individuals. There may exist a poverty environmental trap. That is to say, the environmental deterioration injures health and aggravates the risk of disease, which will lead to increased medical expenditure. Specifically, the work of low-income groups is mostly outdoor work exposed to the air, while the work of high-income groups is mostly indoor work. The low-income groups will face more serious environmental damage and higher medical expenditure, which will exacerbate income inequality [29]. The theory of diminishing marginal propensity to consume points out that the expansion of the income gap will reduce the total consumption of society as a whole [30]. On the other hand, air pollution reduces residents’ expectations of the future, thus increasing savings and reducing purchasing power. The life cycle hypothesis points out that the consumption of residents depends not only on the income of the current period, but also on the expectation of the future. Air pollution distorts wages and increases unemployment, which makes mankind have a bad expectation for the future [29,31]. In order to maximize the utility of the whole life cycle, residents will choose to increase current savings and reduce consumption activities [32]. The decline in residents’ purchasing power shrinks the market scale [33]. Market demand is a prerequisite for determining market supply. According to new economic geography, enterprises tend to cluster in regions with a large market scale [34]. Enterprises will gain more benefits in the process of market scale expansion, because market scale expansion means that the “economic cake” becomes larger. On the contrary, the reduction of market scale will make enterprises face less market demand, sell fewer goods and obtain fewer profits. Many enterprises transfer or go bankrupt with the decrease of profits, which reduces the number of enterprises in a region. Therefore, it can be inferred that air pollution inhibits the site selection of enterprises by reducing the regional market scale. According to the above analysis, we propose Hypothesis 1 and Hypothesis 2 as follows.

**Hypothesis** **1.**
*Air pollution inhibits the site selection of enterprises.*


**Hypothesis** **2.**
*Air pollution inhibits the siting of enterprises by reducing labor endowment and shrinking market scale.*


### 2.2. The Heterogeneity Effect of Air Pollution on Enterprises’ Site Selection

Enterprises of different scales and cleanliness may have different responses to air pollution, resulting in different types of enterprises with different site choices. From the perspective of enterprises’ scale size, enterprises of different scale size have different preferences for factor costs, factor endowments, market scale and industrial policies in the process of site selection. When facing air pollution, there will be differences in their willingness to re-select locations. Specifically, compared with large-scale enterprises, small-scale enterprises have small production and operation scales, single product category, and poor market expansion ability, and therefore additional environmental costs caused by air pollution may have a greater impact on the site selection of small-scale enterprises. From the perspective of enterprises’ cleanliness, the labor force in low-cleaning enterprises is often directly exposed to serious air pollution, which makes them more prone to respiratory diseases, emotional anxiety and other problems [35]. In order to prevent the loss of labor, the subsidies for air pollution of low-cleaning enterprises will be higher. In addition, the labor force often chooses to live near the workplace, and the health of their families (including children and the elderly) is also more vulnerable to the threat of air pollution [36,37]. The possibility of illness is also greater, which makes the adult labor force spend more time and energy taking care of their families, reducing work efficiency. Therefore, low-cleaning enterprises may be more reluctant to build in areas with high air pollution. According to the above analysis, we propose Hypothesis 3 as follows.

**Hypothesis** **3.**
*The impact of air pollution on the site selection of enterprises depends on the scale and cleanliness of enterprises.*


## 3. Methodology and Data

### 3.1. Spatial Econometrics Model

Tobler’s First Law points out that everything is related to other things, and shorter distance things are more related to each other [38]. The theory implies that nothing exists in isolation, and ignoring the spatial correlations in an econometric analysis when variables have spatial correlation results in biased estimator and incorrect inferences [39]. It is worth noting that the spatial relationships inherited in the data will lead to potential spatial autocorrelation problems [40,41]. Thus, in order to overcome the above problem, we should adopt spatial econometric technology to explore the impact of air pollution on the site selection of enterprises. At present, there are three common spatial econometric models: the spatial autoregressive model (SAR), the spatial error model (SEM), and the spatial Durbin model (SDM). The SDM is much more favored by the researchers than both the SAR and the SEM, and should be given priority in empirical analysis [15]. The main reason is that the SDM subsumes the SAR and the SEM, and the SDM can reflect the spatial dependence caused by explained variables, explanatory variables and error terms well [42,43]. The coefficients of SDM still remain unbiased even if SAR or SEM are adopted for real data [15]. Therefore, we consider the SDM model preferentially in our analysis [15,42]. Referring to the relevant scholars, the Benchmark regression model is constructed as follows [15,44]:(1)$$site_{it}=\alpha_{0}+\rho W_{ij}site_{jt}+\alpha_{1}pm_{it}+\alpha_{2}W_{ij}pm_{jt}+\sum_{k=1}^{K}\beta_{k}X_{kit}+\sum_{k=1}^{K}W_{ij}X_{kjt}\theta_{k}+\mu_{i}+\mu_{t}+\varepsilon_{it}$$
where subscripts i and t denote prefecture-level cities and years; variable site refers to enterprises’ site selection; pm represents air pollution levels; W is the spatial weights matrix; X refers to the control variables; $\mu_{i}$ is the city fixed effect; $\mu_{t}$ is the year fixed effect; $\varepsilon_{it}$ is the random error term; $\alpha_{0}$, $\alpha_{1}$, $\alpha_{2}$, ρ, $\beta_{k}$ and $\theta_{k}$ are the corresponding parameters.

### 3.2. Variables

(1) The explained variable: enterprises’ site selection (site). Referring to the relevant scholars, the explained variable site is represented by the logarithmic values of the number of manufacturing enterprises for each city at the end of the year [45]. In order to analyze the heterogeneity of the cleanliness of enterprises, this paper calculates the logarithm of the number of high-cleaning enterprises and low-cleaning enterprises in each city as the explained variable after dividing the enterprises according to the pollution cleanliness. At the same time, enterprises are also divided into large-scale enterprises and small-scale enterprises according to the differences in enterprises’ scale size.

(2) The core explanatory variable: air pollution (pm). Considering that the harmfulness of PM2.5 and PM2.5 concentrations has become one of the most concerning air-quality indicators in recent years, the paper selects the annual average PM2.5 concentration of each region as the core explanatory variable, which is denoted as pm.

(3) Control variables. Following the practice of most scholars, informatization level, government intervention degree, educational level, financing convenience degree, and fixed assets scale are selected as the control variables. The control variables are defined as follows [46].

① Informatization level (web) is characterized by the internet penetration rate, which is disclosed by the China Internet Network Information Center. ② Government intervention degree (gov) is estimated by the ratio of government expenditure to GDP in each region. ③ Educational level (edu) is calculated by the ratio of the number of ordinary university students to the total population at the end of the year in each region. ④ Financing convenience degree (fc) is estimated by the ratio of various loan balances of financial institutions to GDP at the end of each year. ⑤ Fixed assets scale (fix) is expressed by the ratio of total fixed assets investment to GDP in each region.

(4) Weight matrix. Constructing a suitable spatial weight matrix is the premise of using spatial econometric methods. The spatial correlation between regions is not only affected by geographical distance, but also restricted by economic behavior. Meanwhile, the weight matrix constructed only by geographical distance or economic difference cannot correctly reflect the complex relationship between regions. To obtain more accurate results, this paper integrates the geographical and economic factors to build the hybrid economic and distance weighted matrix. The construction method of the hybrid weighted matrix is as follows:(2)$$W_{ij}=\left\{\begin{array}{cc} (\bar{Q_{i}}\times \bar{Q_{j}})/d_{ij}^{2} & i\neq j\\ 0 & i=j \end{array}\right.$$
where $d_{ij}$ denotes the geographical distance between city i and city j, which is calculated by the coordinates of two cities. $\bar{Q_{i}}$ and $\bar{Q_{j}}$ represent the real GDP per capita of the two corresponding cities, respectively. Besides that, this paper obtains the coordinates of each city through Baidu API, and the per capita GDP comes from the China Urban Statistical Yearbook. It is evident that the hybrid economic and distance weighted matrix can reflect the reality and obtain accurate empirical results more accurately.

### 3.3. Data

This paper uses the panel data of 288 cities in China from 2014 to 2020 as the research sample. The data of the explained variable are derived from the Tianyancha Enterprises Information database, which contains the information of all manufacturing enterprises in China. The data of core explanatory variable and control variables are collected from the China Statistical Yearbook, China Urban Statistical Yearbook, and CSMAR database. In addition, for a few missing values, the polynomial fitting method is used to interpolate and fill. Descriptive statistics of all values are shown in Table 1.

## 4. Results

### 4.1. Spatial Correlation Test

This paper employs Moran’s I index to judge whether air pollution and enterprises’ site selection exhibit spatial correlation. The Moran’s I test results based on the hybrid matrix for each year from 2014 to 2020 are displayed in Table 2. It can be seen that the Moran’s I index of air pollution and enterprises’ site selection are greater than zero at a significant level of 1%, which indicates that both air pollution and enterprises’ site selection show a positive spatial dependence. The Moran’s I scatter plots are exhibited in Figure 1 and Figure 2. It is easy to see that the spots in the figures are mostly centralized in quadrants I and III, indicating that there exists spatial clustering. The spatial distribution of enterprises and air pollution is shown in Figure 3 and Figure 4. Specifically, firstly, the enterprises’ site selection indicator is calculated by the arithmetic mean values of the enterprises’ site selection (2014–2020) and is depicted in Figure 3. It is easy to see that different colors represent the distribution of enterprises in different regions: the cyan area indicates that the indicator is in the range of [5.231231, 7.025507], denoting the first level; the blue area indicates that the indicator is in the range of [7.025508–7.936190], representing the second level; the purple area indicates that the indicator is in the range of [7.936191–9.168070], standing for the third level; the red area indicates that the indicator is in the range of [9.168071–11.473110], signifying the highest level; the white area indicates that there are no data in this area. It also can be seen that the number of manufacturing enterprises in China is decreasing from east to west. The eastern and southern coastal areas of China tend to have a higher degree of enterprises agglomeration. Because the adjacent areas have similar economic, financial and educational levels, and industrial policies, together with scale effects, technological externalities, and other reasons, enterprises in the adjacent areas tend to show similar agglomeration, so the siting of enterprises has the characteristics of spatial agglomeration. Secondly, the air pollution indicator, which is calculated by the arithmetic mean values of the air pollution index (2004–2020), is pictured in Figure 4. Similar to Figure 3, different colors represent the degree of air pollution in different regions. The green area indicates that the indicator is in the range of [2.683361–3.407261], denoting the first level; the light green area indicates that the indicator is in the range of [3.407262–3.711487], representing the second level; the light blue color area indicates that the indicator is in the range of [3.711488–4.009135], signifying the third level; the blue area indicates that the indicator is in the range of [4.009136–4.398143], standing for the highest level; the white area indicates that there are no data in this area. It can be seen that air pollution in China is decreasing from the middle to the north and south. The heavily polluted areas are mainly concentrated in Hebei, Shandong and Henan, and other provinces. Air pollution depends not only on the emission of pollutants in a city, but also on the geographical factors to a large extent, and exhibits a strong spillover effect. The large amount of pollutants in a high-pollution city can easily spread to surrounding cities, reducing the air quality of surrounding cities. Therefore, air pollution also has the characteristics of spatial agglomeration. Finally, as previously analyzed, air pollution changes the cost and profit of enterprises by reducing the local labor endowment and market scale, further affecting the location of enterprises. Due to the constant flow of elements, local air pollution changes the distribution of elements in adjacent areas and further affects the site selection of enterprises. To sum up, the impact of air pollution on the siting of enterprises has a strong spatial effect, which cannot be captured by using the traditional linear models. Therefore, this paper uses the spatial econometric model to analyze the impact of air pollution on enterprises’ site selection.

### 4.2. The Baseline Results

Selecting the optimal spatial econometric model is helpful to investigate precisely the reasons for spatial dependence and the impact of spatial correlation mechanisms. Hence, we carry out a series of tests to decide which model is the supreme one to describe the data. Table 3 presents the relevant test results. First, it is imperative to check the spatial effects. Corresponding results of LM and robust LM tests suggest that there exists strong evidence to reject the null hypothesis of nonexistence spatial effects (accept the alternative hypothesis of existence spatial effects) at the significance level of 1%, which provides additional evidence for the existence of spatial dependence. The Wald and LR tests are conducted subsequently to decide whether the SDM degrades into the SAR or SEM. The null hypothesis is the SDM degrades into the SAR or SEM, and, oppositely, the alternative hypothesis is the SAR or SEM outperforms the SDM. The results of the Wald and LR tests show that null hypothesis is rejected at the significance level of 1%, indicating that the SDM outperforms the other two models and it is necessary to employ the SDM in our analysis. Then, we conduct the Hausman test to determine whether the random effects model outperforms the fixed effects model. The results demonstrate that the fixed effect model is superior to the random effect model (125.59, *p* < 0.01). Finally, the LR tests are conducted to check whether the city-fixed effects and time-fixed effects are significant. The results demonstrate that we can reject the null hypothesis of the joint insignificance of the city-fixed effects (170.44, *p* < 0.01) and the joint insignificance of the time-fixed effects (5881.49, *p* < 0.01), that is, accept the alternative hypothesis of the joint significance of the city-fixed effects and time-fixed effects. These results indicate that it is rational to select a model with city- and time- fixed effects. Therefore, we employ the SDM model with spatial and time- fixed effects to conduct our research.

Significantly, the estimation results based on the OLS method may cause bias due to the spatial lag term of enterprises’ site selection contained in Equation (1) [39]. For this reason, following the previous literature, we apply the MLE to estimate our model [15]. Table 4 reports the regression results of the spatial Durbin model. What draws our attention is that the estimated coefficient of PM2.5 is highly significant and negative, which indicates that air pollution reduces the site selection of enterprises, and a city with more serious air pollution decreases the willingness of enterprises to select a site in that city. The parameter ρ is significantly positive, which illustrates that there is an obvious spatial dependence on the site selection of enterprises in various cities. It suggests that enterprises’ site selection in adjacent cities exerts a positive impact on the site selection of enterprises in local cities. The coefficient of the informatization level is statistically significant and positive, indicating that a higher level of informatization contributes to a greater number of enterprises. The coefficient of education level states that a city with higher education levels will have an increased number of enterprises, the reason being that education can improve labor productivity and promotes the development of enterprises. The estimation coefficient of financing convenience degree is positive at the 10% level, implying that the enhancement of financing convenience degree decreases the financing cost and financing threshold of enterprises, which is conducive to enterprises’ site selection. However, the coefficients of the government intervention degree and fixed assets scale are not significant, indicating that the government intervention degree and fixed assets scale are not the main influencing factors in enterprises’ site selection.

The above estimation of the SDM model provides some interesting results; however, the coefficients of the explanatory variables cannot indicate the marginal impact of the explanatory variables on the local explained variable [47]. On the basis of Table 4, this paper further performs partial differential decomposition on the regression results to obtain the direct and indirect effects of explanatory variables. Specifically, the direct effects contain two parts: one part is the direct influence of the explanatory variables on the local explained variable; the other part is the feedback effect, which is that the local explanatory variable affects the explained variables in the nearby cities, and then it conversely influences the local explanatory variables. Indirect effects represent the impact of the explanatory variables of nearby cities on the local enterprises’ site selection, or the influence of alterations in explanatory variables of a local city on the enterprises’ site selection of nearby cities. Obviously, the total effects are the sum of the direct and spillover effects. As revealed in Table 5, the direct effect, spillover effect, and total effect values of the core explanatory air pollution variable are significantly negative and equal −0.115, −0.911 and −1.026, which indicates that the deterioration of local air quality does not only restrain the local enterprises’ site selection, but also prevent the enterprises from locating in neighboring cities. The above results verify Hypothesis 1. In addition, the estimated value of the air pollution spillover effect is much larger than that of the direct effect. The main reason is as follows: on the one hand, spatial spillovers denote the total of spillovers from all neighboring cities. On the other hand, air pollution not only depends on the emission of pollutants in a city, but also is affected by geographical factors to a large extent. For example, basin cities are more unfavorable to the diffusion of pollutants, and coastal cities tend to have better air quality. At the same time, pollutants have a strong spillover effect. A city has a large amount of pollutant emissions can easily spread to surrounding cities. The air quality monitoring stations, which are used to detect regional air quality trends and can radiate 500 m to tens of kilometers, are often shared by neighboring cities. This also indicates that the air pollution levels of neighboring cities are similar, that is, the air pollution exhibits spatial spillover effects in neighboring cities.

### 4.3. The Robustness Test

Several robustness tests are performed in our study to check the stability of the estimation results. Firstly, we employ a replaceable measurement method for air pollution to inspect whether the research conclusions will change. Secondly, the additional weight matrix is employed to test the robustness of the results. Lastly, we apply the lag value of air pollution to solve the possible potential endogenous problem of air pollution.

#### 4.3.1. Replace Core Explanatory Variables

Considering that the estimation results of the impact of air pollution on the site selection of enterprises may vary with different air pollution indicators, we consider replacing the measurement method of air pollution. To that end, according to the research of Yu et al., the air quality index (aqi) is served as a substitute indicator for air pollution, which comprehensively considers the average concentrations of SO_2_, NO_2_, PM10, PM2.5, O_3_ and CO and is more reliable [48]. Since the air quality index of each city published in the CNRDS database is daily data, and the empirical analysis takes the annual data as the research object, this paper takes the average of the daily air quality index data of each city as the annual data. As can be seen from Table 6, the direct effect, spillover effect and total effect of aqi are basically consistent with the results of PM2.5. Our main results are basically unchanged, which indicates that our conclusion is reliable.

#### 4.3.2. Different Spatial Weight Matrix

The robustness check is conducted again by using a different spatial weight matrix. This paper employs the inverse distance weight matrix to carry out the research again. To be specific, the weight matrix meets $w_{ij}=1/d_{ij}$ when city i is not equal to city j, otherwise, we set $w_{ij}=0$. The regression results in Table 7 indicate that using the inverse distance weight matrix does not alter our main conclusions.

#### 4.3.3. Endogenous Issue

The impact of air pollution on enterprises is usually a long-term cumulative process, which may have a certain lag. So referring to the method of relevant scholars, we use a first-order lagged value of air pollution as a proxy for air pollution to mitigate the underlying estimation error generated by the reverse causality between air pollution and enterprises’ site selection [49]. The estimation results are displayed in Table 8. The results indicate that the coefficients of the core explanatory variable are approximately the same as the baseline regression, which confirms our conclusions once again.

### 4.4. Heterogeneity Analysis

Different enterprises may have different responses to air pollution due to different characteristics, which may result in different regression results. Considering the difference of enterprises’ cleanliness and scale size, this paper further conducts heterogeneity analysis.

#### 4.4.1. Heterogeneity of Enterprises Cleanliness

The site selection of enterprises with different cleanliness may be affected by air pollution differently. Therefore, referring to the classified management directory of environmental protection verification industry of Listed Companies, formulated by the Ministry of environmental protection of China in 2008, this paper classifies the sample enterprises into high-cleaning enterprises and low-cleaning enterprises. The regression results are shown in Table 9. Under the condition that other factors remain unchanged, the direct effect, spillover effect and total effect of air pollution on the site selection of both high- and low-cleaning enterprises are negative and highly significant. Meanwhile, the absolute values of direct effect regarding air pollution on the site selection of low-cleaning enterprises and high-cleaning enterprises do not exhibit much difference, while the absolute values of spillover effect and total effect of the former are greater than the latter. Serious air pollution in a city is often accompanied by higher environmental standards. There exists competitive behavior among regions due to promotion incentives and other factors, which results in a region that refers to and formulate policies that are higher than the environmental standards of other regions [50]. As a result, all regions compete to raise the entry threshold of low-cleaning enterprises. Compared with high-cleaning enterprises, low-cleaning enterprises bear greater environmental responsibility and pay higher environmental protection costs. However, because most employees in low-cleaning enterprises work outdoors, air pollution does more harm to their health. Low-cleaning enterprises have higher requirements for environmental quality. Therefore, air pollution has a stronger inhibitory effect on the site selection of low-cleaning enterprises.

#### 4.4.2. Heterogeneity of Enterprises’ Scale

The impact of air pollution on the site selection of enterprises with different scale sizes may be different. Therefore, referring to the definition method of the China Industrial Statistics Yearbook, this paper divides all sample enterprises into large-scale enterprises and small-scale enterprises according to the enterprises’ registered capital index. Specifically, enterprises with a registered capital of less than 5 million yuan are defined as small-scale enterprises, and enterprises with a registered capital of more than 5 million yuan (including 5 million yuan) are defined as large-scale enterprises. The regression results are shown in Table 10. It can be seen that the direct effect, spillover effect and total effect of large-scale enterprises are significantly negative, while the direct effect on small-scale enterprises is not significant, and the indirect effect and total effect are significantly negative. Meanwhile, the absolute values of spillover effect and total effect of large-scale enterprises is smaller than that of small-scale enterprises. Generally speaking, large-scale enterprises have higher resource endowments and financial resources than small-scale enterprises, and have a stronger ability to cope with external shocks such as air pollution. For example, large-scale enterprises have enough capacity to pay pollution charges and compensate for the damage to employees due to air pollution. In addition, large-scale enterprises occupy a stable market share in the local area, have more employees and fixed assets, and the cost of relocation is higher. Large-scale enterprises are more tolerant of the negative impact of air pollution. Therefore, air pollution has a stronger inhibitory effect on the site selection of small-scale enterprises.

Therefore, the conclusions of the above two kinds of heterogeneity analysis both validate Hypothesis 3.

### 4.5. Mechanism Analysis

The previous sections confirm that air pollution has a negative effect on the site selection of enterprises. Therefore, we perform several mechanism analyses to explore the latent interprets for the above results in this section. We construct a classical mediating effect model as follows.
(3)$$site_{it}=\alpha_{0}+\rho W_{ij}site_{jt}+\alpha_{1}pm_{it}+\alpha_{2}W_{ij}pm_{jt}+\sum_{k=1}^{K}\beta_{k}X_{kit}+\sum_{k=1}^{K}W_{ij}X_{kjt}\theta_{k}+\mu_{i}+\mu_{t}+\varepsilon_{it}$$
(4)$$med_{it}=\alpha_{0}+\rho W_{ij}med_{jt}+\alpha_{1}pm_{it}+\alpha_{2}W_{ij}pm_{jt}+\sum_{k=1}^{K}\beta_{k}X_{kit}+\sum_{k=1}^{K}W_{ij}X_{kjt}\theta_{k}+\pi_{i}+\pi_{t}+\varepsilon_{it}$$
(5)$$\begin{array}{ll} site_{it}= & \alpha_{0}+\rho W_{ij}site_{jt}+\gamma_{1}med_{it}+\gamma_{2}W_{ij}med_{jt}+\alpha_{1}pm_{it}+\alpha_{2}W_{ij}pm_{jt}+\\ & \sum_{k=1}^{K}\beta_{k}X_{kit}+\sum_{k=1}^{K}W_{ij}X_{kjt}\theta_{k}+\tau_{i}+\tau_{t}+\varepsilon_{it} \end{array}$$
where, med denotes intermediary variable, representing labor endowment and market scale, respectively, both of which are obtained from the China Urban Statistical Yearbook. Formula (3) is the same as Formula (1), and all the other variables have the same meaning as Formula (1) and possess the same source of data as above.

#### 4.5.1. The Effect of Reduce Labor Endowment

To verify whether air pollution changes the location of enterprises by affecting labor endowment, we take labor endowment (labor) measured by the logarithm of employment in each city as the intermediary variable, and the estimation results of Equations (4) and (5) with the spatial weight matrix of W are reported in Table 11. Columns (1)–(3) of Table 11 display the regression results of air pollution on labor endowment. The estimation results indicate that the direct, indirect and total effects values of air pollution on labor endowment are negative and significant. The results illustrate that air pollution reduces a region’s labor endowment, that is, air pollution damages the physical and mental health of residents, causing a shortness of labor supply, transfer or death. Columns (4)–(6) show the regression results of air pollution and labor endowment on the site selection of enterprises. The direct, indirect and total effects values of labor endowment on enterprises’ site selection are positive and significant. The above results indicate that air pollution inhibits the site selection of enterprises by reducing labor endowment.

#### 4.5.2. The Effect of Shrinking Market Scale

Similarly, we take market scale (mar), measured by the ratio of total retail sales of social consumer goods to the total population in each region, as the intermediary variable, and the estimation results of Equations (4) and (5) with the spatial weight matrix of W are displayed in Table 12. Columns (1)–(3) display the regression results of air pollution on market scale. The estimation results indicate that the direct, indirect and total effects values of air pollution on market scale are negative and significant, which implies that air pollution reduces the market scale by reducing residents’ consumption and income. Columns (4)–(6) show the regression results of air pollution and market scale on the site selection of enterprises. The direct, indirect and total effects values of market scale on enterprises’ site selection are positive and significant. The above results confirm that air pollution inhibits the site selection of enterprises by shrinking market scale.

In short, the conclusions of the above two kinds of mechanism analysis both verify Hypothesis 2.

## 5. Conclusions

In consideration of the growing concerns of global climatic change, air pollution has developed into one of the most important factors affecting the site selection of enterprises. However, few scholars have investigated how enterprises change their location to cope with air pollution, let alone the spatial spillover effects of air pollution on enterprises’ site selection. Based on the prefecture-level panel data of China during the period 2014–2020, this paper applies the spatial Durbin model to investigate the impact of air pollution on enterprises’ site selection. The empirical results show that there is a significant spatial spillover effect of local air pollution on enterprises’ site selection in neighboring cities. Concretely, air pollution reduces enterprises not only locally but also in spatially related regions. Considering the differences of enterprises’ scale size and enterprises’ cleanliness, the results of heterogeneity analysis suggest that air pollution has a greater inhibitory effect on the site selection of low-cleaning enterprises and small-scale enterprises in local and adjacent regions. In addition, the results of mechanism analysis demonstrate that air pollution mainly inhibits enterprises’ site selection by reducing labor endowment and market scale.

Based on the above conclusions, our study can obtain the following implications. The government should be aware of the importance of improving environmental quality to attract enterprises and promote economic development. Obviously, this paper confirms the inhibitory effect of air pollution on the site selection of local and spatially related regional enterprises. Due to many environmental problems accumulated in history, the problems of unbalanced, uncoordinated and unsustainable development are still prominent. As a result, China is still not optimistic regarding its air pollution problems. Thus the government should strengthen environmental governance, taking improving environmental quality as the core, and promote the implementation of major environmental protection actions. In addition, the government should increase financial input and vigorously promote the production and use of clean energy. For the public, with the development of the economy and the enrichment of material life, they are increasingly concerned about their own health and the surrounding environment. The environmental quality has an impact on people’s life and consumption. Therefore, the government should encourage grassroots people to establish autonomous organizations, social organizations and environmental protection volunteers to publicize environmental protection laws and regulations, and environmental protection knowledge, and create a good atmosphere for environmental protection. Furthermore, the government also needs to increase support for residents’ medical security, eliminate their worries and improve their quality of life. For enterprises, air pollution has a greater inhibitory effect on small-scale enterprises and low-cleaning enterprises’ site selection. The state and relevant institutions should devote themselves to exploring methods and paths that can fundamentally improve China’s environmental pollution problem, and guide small-scale enterprises and low-cleaning enterprises to improve their traditional production modes of high energy consumption, high pollution and low output. In addition to the use of mandatory means, the government needs to reinforce the education and publicity of enterprises to make them aware that the air pollution they create will conversely inhibit their business performance and sustainable development.

## Figures and Tables

**Figure 1 ijerph-19-14484-f001:**
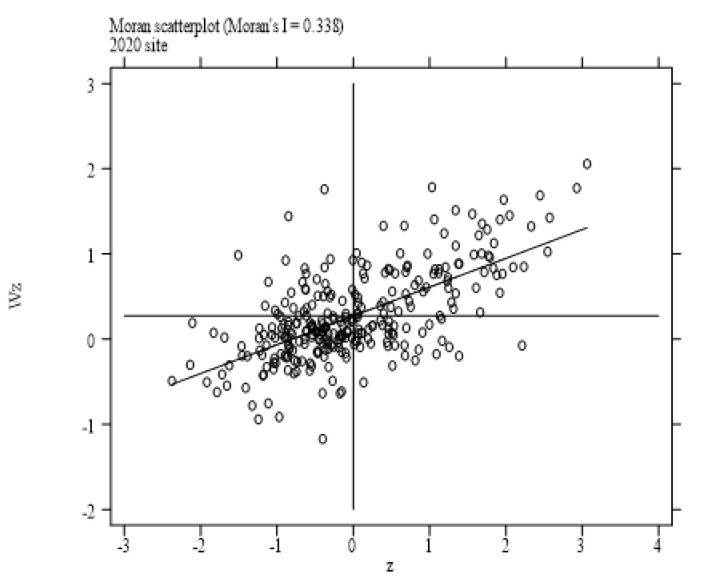
Moran’s scatter plot of enterprises’ site selection.

**Figure 2 ijerph-19-14484-f002:**
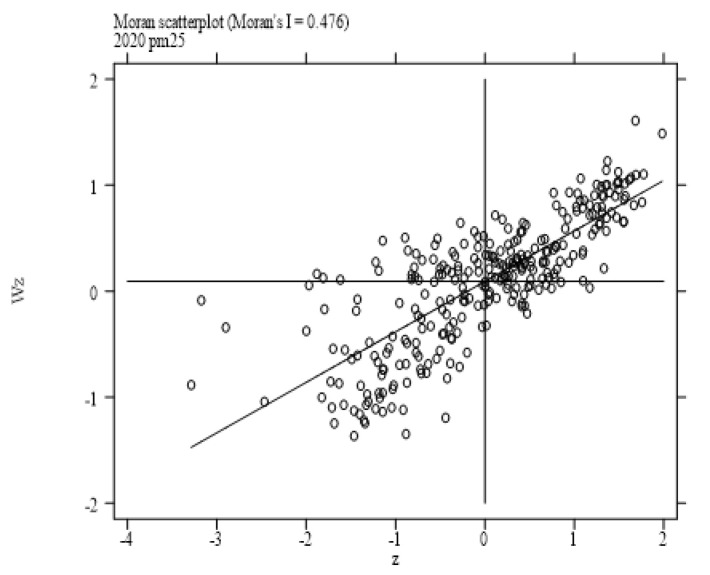
Moran’s scatter plot of air pollution.

**Figure 3 ijerph-19-14484-f003:**
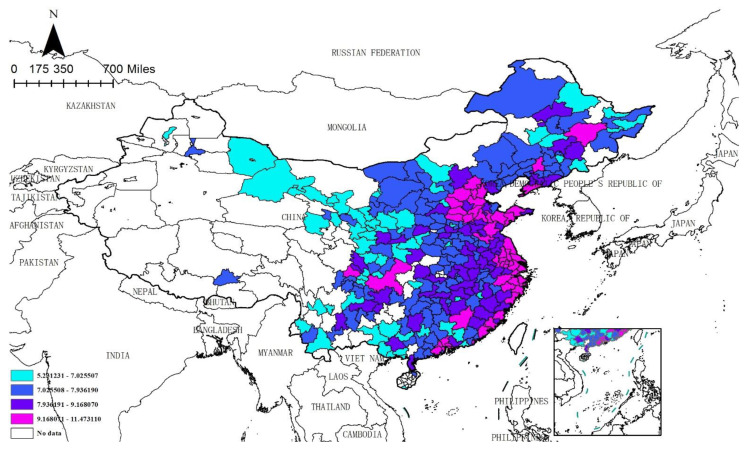
Enterprises Distribution. Note: The arithmetic mean values of the enterprises’ site selection (2014–2020) is adopted.

**Figure 4 ijerph-19-14484-f004:**
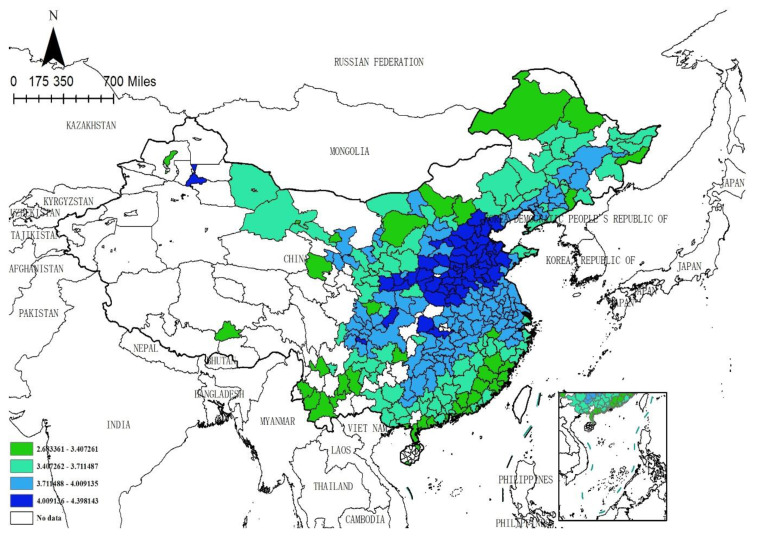
Air pollution distribution. Note: The arithmetic mean values of the air pollution index (2004–2020) is adopted.

**Table 1 ijerph-19-14484-t001:** Descriptive statistics.

Variables	Unit	Obs	Mean	Std. Dev.	Min	Max
site	a	2016	7.934	1.272	3.434	11.849
pm	$\mu g/m^{3}$	2016	3.74	0.395	2.197	5.17
web	%	2016	0.583	0.072	0.347	0.831
gov	%	2016	0.219	0.129	0.023	2.06
edu	%	2016	0.019	0.022	0.001	0.118
fc	%	2016	1.143	0.737	0.115	9.622
fix	%	2016	1.291	1.054	0.182	4.917

**Table 2 ijerph-19-14484-t002:** Statistical tests of spatial autocorrelation by Moran’s I.

	2014	2015	2016	2017	2018	2019	2020
site	0.411 ***	0.391 ***	0.376 ***	0.364 ***	0.356 ***	0.348 ***	0.338 ***
pm	0.170 ***	0.474 ***	0.458 ***	0.382 ***	0.426 ***	0.438 ***	0.476 ***

Notes: *** denote significance of 1%.

**Table 3 ijerph-19-14484-t003:** Applicability test of spatial panel model.

	$\chi^{2}$	*p* Values
LM spatial lag	2038.003	0.0000
Robust LM spatial lag	158.162	0.0000
LM spatial error	3183.308	0.0000
Robust LM spatial error	1303.467	0.0000
LR test spatial lag	171.35	0.0000
Wald test spatial lag	30.79	0.0000
LR test spatial error	188.45	0.0000
Wald test spatial error	22.99	0.0008
Hausman test	125.59	0.0000
LR test for ind	170.44	0.0010
LR test for time	5881.49	0.0000

**Table 4 ijerph-19-14484-t004:** Result of spatial Durbin models.

Variables	Coefficient	Std. Err
pm	−0.138 ***	0.0204
web	0.359 ***	0.123
gov	0.0110	0.0561
edu	0.462 *	0.248
fc	0.0166 *	0.00979
fix	0.00485	0.00681
W*pm	0.345 ***	0.0460
W*web	−0.633 ***	0.228
W*gov	−0.340	0.369
W*edu	2.009 ***	0.752
W*fina	−0.113 **	0.0494
W*fc	−0.0295 *	0.0164
ρ	1.202 ***	0.0110
*N*	1728	
*R* ^2^	0.045	

Notes: ***, ** and * denote significance of 1%, 5% and 10%, respectively.

**Table 5 ijerph-19-14484-t005:** Direct, indirect and total effects for Table 4.

Variables	Direct Effect		Indirect Effect		Total Effect	
	Coefficients	Std. Err	Coefficients	Std. Err	Coefficients	Std. Err
pm	−0.115 ***	0.0227	−0.911 ***	0.181	−1.026 ***	0.176
web	0.328 **	0.127	1.065	0.998	1.393	0.954
gov	−0.0177	0.0706	1.492	1.826	1.475	1.787
edu	0.771 **	0.300	−13.15 ***	4.091	−12.38 ***	3.949
fc	0.00469	0.0119	0.493 *	0.252	0.498 **	0.246
fix	0.00228	0.00686	0.117 *	0.0700	0.119 *	0.0691

Notes: ***, ** and * denote significance of 1%, 5% and 10%, respectively.

**Table 6 ijerph-19-14484-t006:** Robust test with the alternative core explanatory variables.

Variables	Direct Effects		Spillover Effects		Total Effects	
	Coefficients	Std. Err	Coefficients	Std. Err	Coefficients	Std. Err
aqi	−0.182 ***	0.0370	−0.972 ***	0.369	−1.154 ***	0.368
web	0.334 ***	0.124	1.278	1.089	1.612	1.057
gov	0.0197	0.0598	−0.265	1.950	−0.246	1.928
edu	0.667 **	0.288	−17.94 ***	4.314	−17.27 ***	4.214
fc	0.00139	0.0116	0.960 ***	0.251	0.962 ***	0.247
fix	0.00227	0.00682	0.168 **	0.0751	0.170 **	0.0749

Notes: *** and ** denote significance of 1% and 5%, respectively.

**Table 7 ijerph-19-14484-t007:** Robust test using a different weight matrix.

Variables	Direct Effects		Spillover Effects		Total Effects	
	Coefficients	Std. Err	Coefficients	Std. Err	Coefficients	Std. Err
pm	−0.145 ***	0.0305	−1.917 ***	0.502	−2.062 ***	0.507
web	0.506 ***	0.192	3.266 *	1.882	3.772 **	1.886
gov	−0.0123	0.0929	0.998	2.770	0.985	2.783
edu	1.114 **	0.433	8.341	16.20	9.454	16.31
fc	−0.000629	0.0163	0.0114	0.534	0.0107	0.536
fix	0.00877	0.00976	0.124	0.0805	0.133	0.0813

Notes: ***, ** and * denote significance of 1%, 5% and 10%, respectively.

**Table 8 ijerph-19-14484-t008:** Robust test with the second-order lagged value of air pollution.

Variables	Direct Effects		Spillover Effects		Total Effects	
	Coefficients	Std. Err	Coefficients	Std. Err	Coefficients	Std. Err
pm	−0.0482 **	0.0205	−0.712 ***	0.146	−0.761 ***	0.144
web	0.191 **	0.0746	−0.526	0.679	−0.335	0.670
gov	0.0785	0.0593	−0.151	1.560	−0.0722	1.560
edu	0.342	0.311	−11.20 ***	2.875	−10.86 ***	2.825
fc	−0.00919	0.00974	0.199	0.205	0.189	0.205
fix	−0.00109	0.00733	0.153 ***	0.0551	0.151 ***	0.0558

Notes: *** and ** denote significance of 1% and 5%, respectively.

**Table 9 ijerph-19-14484-t009:** Results based on heterogeneity of enterprises’ pollution degree.

Variables	Low-Cleaning Enterprises			High-Cleaning Enterprises		
	Direct Effects	Spillover Effects	Total Effects	Direct Effects	Spillover Effects	Total Effects
pm	−0.115 *** (0.0278)	−1.025 *** (0.233)	−1.141 *** (0.228)	−0.116 *** (0.0224)	−0.902 *** (0.181)	−1.018 *** (0.176)
web	0.322 ** (0.158)	1.488 (1.296)	1.810 (1.247)	0.352 *** (0.126)	1.148 (0.995)	1.501 (0.953)
gov	−0.0529 (0.0819)	0.815 (2.373)	0.762 (2.334)	−0.0121 (0.0691)	1.574 (1.823)	1.562 (1.785)
edu	0.439 (0.354)	−10.26 * (5.288)	−9.823 * (5.121)	0.835 *** (0.297)	−14.03 *** (4.090)	−13.19 *** (3.955)
fc	0.00123 (0.0142)	0.600 * (0.328)	0.601 * (0.321)	0.00603 (0.0116)	0.470 * (0.251)	0.476 * (0.245)
fix	−0.00505 (0.00851)	0.125 (0.0909)	0.120 (0.0902)	0.00162 (0.00680)	0.124 * (0.0699)	0.126 * (0.0690)

Notes: ***, ** and * denote significance of 1%, 5% and 10%, respectively. Numbers in the () represent standard errors.

**Table 10 ijerph-19-14484-t010:** Results based on heterogeneity of enterprises’ scale size.

Variables	Large Scale Enterprises			Small Scale Enterprises		
	Direct Effects	Spillover Effects	Total Effects	Direct Effects	Spillover Effects	Total Effects
pm	−0.112 *** (0.0231)	−0.893 *** (0.192)	−1.005 *** (0.188)	−0.271 (0.240)	−0.916 *** (0.258)	−1.187 *** (0.104)
web	0.385 *** (0.131)	0.698 (1.054)	1.083 (1.011)	0.427 (0.487)	1.871 ** (0.727)	2.297 *** (0.564)
gov	0.00424 (0.0702)	1.472 (1.927)	1.476 (1.891)	0.0415 (0.309)	0.600 (0.998)	0.641 (1.054)
edu	0.826 *** (0.302)	−12.65 *** (4.318)	−11.82 *** (4.180)	−0.436 (1.173)	−6.108 ** (2.501)	−6.544 *** (2.307)
fc	0.00287 (0.0120)	0.561 ** (0.265)	0.563 ** (0.260)	0.0680 (0.124)	0.438 ** (0.181)	0.506 *** (0.146)
fix	−0.00196 (0.00709)	0.127 * (0.0739)	0.125 * (0.0731)	0.0105 (0.0246)	0.0774 * (0.0424)	0.0880 ** (0.0406)

Notes: ***, ** and * denote significance of 1%, 5% and 10%, respectively. Numbers in the () represent standard errors.

**Table 11 ijerph-19-14484-t011:** Mechanism analysis with labor endowment effect.

Variables	Labor			Site		
	Direct Effects	Spillover Effects	Total Effects	Direct Effects	Spillover Effects	Total Effects
pm	−0.156 *** (0.0302)	−0.469 * (0.284)	−0.626 ** (0.281)	−0.0352 *** (0.0128)	1.156 *** (0.369)	1.121 *** (0.370)
labor				0.546 *** (0.0112)	1.105 *** (0.205)	1.651 *** (0.205)
web	−4.972 *** (0.176)	3.378 ** (1.538)	−1.595 (1.494)	3.076 *** (0.0964)	3.219 *** (1.207)	6.295 *** (1.216)
gov	−0.00889 (0.104)	7.610 *** (2.861)	7.602 *** (2.844)	0.0384 (0.0410)	4.804 ** (2.067)	4.842 ** (2.086)
edu	2.120 *** (0.409)	−24.13 *** (6.400)	−22.01 *** (6.282)	−0.640 *** (0.175)	−1.063 (4.139)	−1.704 (4.220)
fc	0.0267 * (0.0147)	−0.186 (0.390)	−0.160 (0.385)	−0.00910 (0.00766)	−0.855 ** (0.336)	−0.864 ** (0.340)
fix	−0.0136 (0.00957)	0.208 * (0.108)	0.194 * (0.108)	0.0104 ** (0.00456)	0.0269 (0.0698)	0.0373 (0.0695)

Notes: ***, ** and * denote significance of 1%, 5% and 10%, respectively.

**Table 12 ijerph-19-14484-t012:** Mechanism analysis with market scale effect.

Variables	Mar			Site		
	Direct Effects	Spillover Effects	Total Effects	Direct Effects	Spillover Effects	Total Effects
pm	−0.0231 *** (0.00620)	−0.707 * (0.399)	−0.730 * (0.401)	−0.0725 *** (0.0159)	0.882 ** (0.366)	0.810 ** (0.366)
mar				2.203 *** (0.0710)	10.57 *** (1.851)	12.78 *** (1.852)
web	−0.0356 (0.0374)	1.915 (2.003)	1.879 (2.018)	0.441 *** (0.0994)	−1.088 (1.290)	−0.647 (1.309)
gov	−0.0504 * (0.0272)	−4.392 (4.174)	−4.442 (4.196)	0.0751 (0.0500)	0.967 (2.143)	1.042 (2.166)
edu	0.106 (0.0918)	11.26 (9.019)	11.37 (9.076)	0.462 ** (0.219)	6.439 (5.289)	6.901 (5.395)
fc	−0.000547 (0.00406)	0.408 (0.523)	0.407 (0.526)	0.0172 * (0.00941)	−0.435 (0.319)	−0.418 (0.324)
fix	0.0116 *** (0.00196)	−0.0849 (0.135)	−0.0733 (0.135)	−0.0198 *** (0.00571)	−0.0723 (0.0857)	−0.0921 (0.0855)

Notes: ***, ** and * denote significance of 1%, 5% and 10%, respectively.

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
