# Peer review of "Spatial Effects of Air Pollution on the Siting of Enterprises: Evidence from China"

_ijerph, 2022, doi:10.3390/ijerph192114484_

Round 1

Reviewer 1 Report

Dear Authors,

This paper titled: Spatial Effects of Air Pollution on the Siting of Enterprises: Evidence from China, raises a very interesting topic that is not yet described in great detail in the literature. After careful reading I have some comments and advice, which can be useful for the Authors.

Line 23 –  “As the World Health Organization stated … the bad situation would continue for a long time and was likely to get worse” please make citation for that sentence.

Line 32 - labors will choose …  Maybe Authors will find better words instead labors? E.g. employees, staff, workers, decision makers, managers…

Line 96 - Please explain what kind of sentence is this: The effect of reducing labor endowment. Is it subtile? The same question for: The effect of shrinking market scale?

Line 53, Line 182, Line 188 – please make citation according mdpi requirements. LeSage and Pace – there is no in the reference list. Please use [X] style not LeSage and Pace (2009).

 The authors drew attention to health and existential problems caused by air pollution twice. It is worth adding two sentences that air pollution also affects climate change, agriculture, water and the quality of food consumed. The yields may be poorer, etc. This is the nexus of Water-Energy-Food. Please read and used the paper Nexus between water, energy, food and climate change as challenges facing the modern global, European and Polish economy. AIMS Geosci6, 397-421; or/and Air pollution, food production and food security: A review from the perspective of food system. Journal of integrative agriculture16(12), 2945-2962.

Line 197 - Lu Li – similar comments as above. Please check all similar citations.

What does black +_ mean in the Figure 3 and figure 4?

Authors put 3 Hypotheses, but only 1 Hypothesis were verified (line 348). What about the other hypotheses?

Please explain null hypothesis – there are two hypothesis. Null and alternative hypothesis. If the null hypothesis is described, then an alternative hypothesis should also be described. Please explain this.

I think that my comments and advice will be useful for the Authors.

Reviewer 2 Report

Review

Spatial Effects of Air Pollution on the Siting of Enterprises: Evidence from China

 Xu na Zhang, Shi jing Nan *, Shan bing Lu and Min na Wang

Abstract: Lines 6-18

Please structure the Abstract as:

Introduction-Aims

Method

Results and interpretation

Introduction

The aim of the study is not very clear expressed

Literature review should be updated to 2022 and internationalized. Please do an analytical critical review of international and from China related with your topic.

Methods and data

Line 228

The construction method of the hybrid weighted matrix is as follows: please insert the datasource.

Results

Fig. 3 page 7

``Figure 3. Enterprises Distribution. Notes: Note: The arithmetic mean values of the enterprises site  select (2014-2020) is adopted``

Please update data to 2022

Please insert some toponyms, the neighbors

Which is the sense of figure from the small square?

Please insert the units  for the legend values

Fig. 4 line 286

``Figure 4. Air pollution distribution. Note: The arithmetic mean values of air pollution index (2004-2020) is adopted``.

Please update data to 2022

Please insert some toponyms, the neighbors

Which is the sense of figure from the small square?

Please insert the units  for the legend values

For Eq 3,4,5 page 13 please insert datasource if the case

References: internationalize and update the references to 2022

Please cite also:

Marcu, F.; Hodor, N.; Indrie, L.; Dejeu, P.; IlieÈ™, M.; Albu, A.; Sandor, M.; Sicora, C.; Costea, M.; IlieÈ™, D.C.; Caciora, T.; Huniadi, A.; ChiÈ™, I.; Barbu-Tudoran, L.; Szabo-Alexi, P.; Grama, V.; Safarov, B. Microbiological, Health and Comfort Aspects of Indoor Air Quality in a Romanian Historical Wooden Church. Int. J. Environ. Res. Public Health 202118, 9908. https://doi.org/10.3390/ijerph18189908

Is it relevant and
interesting?

The paper is relevant for postpandemic period. It synthetizes the actual available literature data, focus mainly on Improving the green efficiency of siting entreprises and Spatial Effects of Air Pollution in  China Is essential to promoting low-pollution and high-efficient development, and industrial structure is a key factor in this dynamic urban environments.

How original is the topic?

Is an actual topic even it is not very original; important subject in the post pandemic period in China

What does it add to the subject
area compared with other published material?

The paper should be better documented (number of 39 scientific published articles in references list), very few updated to 2022.

Is the paper well written?

The paper is well written. The quality of English translation is good.

Is the text clear and easy to read?

The text is well structured, clear and easy to read from the specialists in the field but as well as from the persons from public.

Are the conclusions consistent with the evidence and arguments presented?

The conclusions are well represented in the paper. The authors underline the potential for improving, in conditions of influencing factors in urban environment milieu, especially for human community. Also are underlined the needs for the government to reinforce the education and publicity of enterprises to make them aware that the air pollution resulted in by themselves will conversely inhibit their business performance and sustainable development.

Round 2

Reviewer 1 Report

Dear Authors, Thank you very much for sending the revised version of your article. All my comments and suggestions have been taken into account, so now the article is well written. Thank you for adding the verified other two research hypotheses. The authors' responses have been very well and carefully compiled. Congratulation.